# Molecular Hydrogen Improves Rice Storage Quality via Alleviating Lipid Deterioration and Maintaining Nutritional Values

**DOI:** 10.3390/plants11192588

**Published:** 2022-09-30

**Authors:** Chenxu Cai, Zhushan Zhao, Yingying Zhang, Min Li, Longna Li, Pengfei Cheng, Wenbiao Shen

**Affiliations:** Laboratory Center of Life Sciences, College of Life Sciences, Nanjing Agricultural University, Nanjing 210095, China

**Keywords:** hydrogen nanobubble water, molecular hydrogen, rice, storage quality, volatile

## Abstract

Improvement of the storage quality of rice is a critical challenge for the scientific community. This study assesses the effects of the irrigation with hydrogen nanobubble water (HNW) on the storage quality of rice (*Oryza sativa* ‘Huruan1212’). Compared with ditch water control, after one year of storage at 25 °C and 70% RH, the HNW-irrigated rice had higher contents of essential amino acids, especially lysine. Importantly, the generation of off-flavors in the stored rice was significantly decreased, which was confirmed by the lower levels of volatile substances, including pentanal, hexanal, heptanal, octanal, 1-octen-3-ol, and 2-heptanone. The subsequent results showed that the HNW-irrigated rice not only retained lower levels of free fatty acid values, but also had increased antioxidant capacity and decreased lipoxygenase activity and transcripts, thus resulting in decreased lipid peroxidation. This study opens a new window for the practical application of HNW irrigation in the production and subsequent storage of crops.

## 1. Introduction

Rice (*Oryza sativa* L.), one of the vital staple foods of the world’s human population in more than 100 countries, is critical for human nutrition, energy supply, and food security [1]. The Food and Agriculture Organization of the United Nations (FAO) reported that the worldwide rice production was 756 million tons in 2020 [2]. Due to the discontinuity of production compared to consumers’ consumption of this staple food, a large proportion of harvested rice normally needs to be stored for a long time. However, official data showed that more than 35 million tons of grains, including rice and wheat, are lost every year in China due to storage, transportation, pest/microbe attacks, etc. [3]. Therefore, the improvement of the storage quality of grains is a critical issue for the scientific community.

Rice flavor is quite complex due to the interactions among numerous volatile compounds, and storage commonly deteriorates rice flavor, resulting in an increase in undesirable volatiles [4]. More than 300 volatiles have been identified in rice, including aldehydes, alcohols, and ketones. Among them, the biosynthesis of some volatiles, including hexanal, heptanal, octanal, pentanal, 1-octen-3-ol, and 2-heptanone, were closely related to microbial activity and lipid deterioration [5,6,7,8]. A portion of these volatiles are the important contributors to the off-flavor of rice during storage [4].

Lipids can be oxidized by two pathways. One is enzymatic-catalyzed oxidation, and the other is autoxidation [9]. In the enzymatic pathway, lipoxygenase (LOX) can catalyze the oxidization of polyunsaturated fatty acids after their release from storage lipids by lipase [10]. Three LOX isozymes, including LOX1, LOX2, and LOX3, were identified in the bran fraction of rice grains, and the rice varieties with LOX1 and LOX2 isozymes were responsible for rice quality deterioration caused by storage [11]. Another study clearly showed that LOX3, the major isozyme component, is mainly responsible for the off-flavor production during storage [9]. In addition, autoxidation is a free radical chain reaction, and its oxidation rate is closely related to a battery of antioxidant molecules and antioxidant enzymes, including superoxide dismutase (SOD), peroxidase (POD), catalase (CAT), and ascorbate peroxidase (APX) [12,13]. 

Hydrogen gas (H_2_), also called molecular hydrogen, is a form of storable and promising clean energy as it is considered to be a potential replacement for fossil fuels in industry. It can be produced by water electrolysis, biomass, and solar energy [14]. Depending on the raw materials and technology, the production cost of H_2_ is in the range of USD 0.8 to 4 per kilogram in different regions. In 2050, hydrogen energy will fulfill approximately 18% of global energy use [15]. It is clear that H_2_ provides a brand-new possibility for the development of a low-carbon society.

Since Japanese scientists discovered that the inhalation of H_2_ can alleviate rat ischemia/reperfusion injuries [16], the clinical value of H_2_ has received increasing attention. Following the progress in medicine, plant scientists were also interested in hydrogen-based agriculture. It has been found that H_2_, when applied either in the form of a gas or a liquid (hydrogen nanobubble water, HNW, etc.), or as a solid hydrogen material (e.g., magnesium hydride; MgH_2_), could extend the storage period of some vegetables and flowers. These include Chinese chives [17], cut carnation flowers [18], etc. More recent results have shown that the positive influence of hydrogen-rich water in butter quality during cold storage was mediated by reducing the formation of biogenic amines [19]. Interestingly, the preharvest application of molecular hydrogen not only enhanced the yield of daylilies, but also abolished the browning level of the buds, thereby prolonging their shelf life [20]. The evaluation of this mechanism further revealed that the positive effects of H_2_ on postharvest preservation were mainly attributed to a broad enhancement of plant antioxidant machinery [21]. However, the elucidation of the biological functions of H_2_ within the food supply chain and the corresponding mechanisms is just beginning. 

We discovered in our previous research that molecular hydrogen improves the qualitative and quantitative traits of rice grain [22]. On this basis, it is necessary to assess the effects of molecular hydrogen for the rice postharvest stage in the food supply chain. Therefore, this study aimed to investigate whether the preharvest application of HNW could improve rice storage quality, and it tried to elucidate the possible corresponding mechanisms. It further proposed a new idea that the application of H_2_ in the crop storage stage might be an efficient and low-carbon approach. 

## 2. Results and Discussion

### 2.1. Preharvest HNW Treatment Alleviates the Generation of Off-Flavor during Rice Storage

Flavor is one of the main rice characteristic quality factors, and it is the result of the relative interaction between specific volatiles and other complex compounds [4]. The volatile compounds of rice include aldehydes, alcohols, ketones, esters, and other compounds [6,23]. Here, a total of 40 major volatiles in brown rice were identified after one year of storage (25 °C, 70% RH), including 11 aldehydes, 8 alcohols, 9 ketones, 5 esters, and 7 other compounds (Appendix A).

During storage, lipid oxidation results in increased contents of odor-active monocarbonyl volatile compounds, especially hexanal, and these compounds have an undesirable impact on the flavor of the stored rice [5,7]. In this study, after HNW irrigation during rice growth and grain filling, the aldehyde content (413.80 ± 26.21 μg kg^−1^) of stored rice was significantly decreased by 15.14% in comparison with the control (487.62 ± 29.33 μg kg^−1^) after one year of storage (Figure 1). However, the preharvest application of HNW had no such obvious effects on the contents of alcohols, ketones, esters, and other volatile compounds, nor on the total volatiles of stored rice.

In particular, the contents of some representative volatile substances which were produced by lipid oxidation were increased markedly with storage time. These included hexanal, heptanal, pentanal, octanal, 1-octen-3-ol, and 2-heptanone [6,24]. Here, we discovered that compared with the control, irrigation with HNW noticeably reduced the contents of hexanal, pentanal, heptanal, octanal, 1-octen-3-ol, and 2-heptanone in brown rice after one year of storage (Figure 2). It was well-known that volatile compounds such as pentanal, hexanal, heptanal, octanal, and 1-octen-3-ol were produced by the autoxidation of oleic acid, linoleic acid, and linolenic acid [7,8]. In this study, the preharvest HNW treatment appeared to reduce the levels of lipid autoxidation products during storage, thus alleviating off-flavor development in stored rice. 

This finding, that the irrigation with HNW improves the flavor quality of stored rice, was consistent with the previous results, showing that the preharvest application of molecular hydrogen improves strawberry flavor by modulating transcriptional profiles [25], and it alleviates daylily bud-browning partially via increasing the unsaturated to saturated fatty acid ratio [20]. Similarly, a more recent result has also shown that the application of hydrogen-rich water during manufacturing could mitigate the formation of biogenic amines in butter during cold storage, likely via decreasing the activity of bacteria decarboxylases or their synthesis [19]. Meanwhile, the generation of 2-heptanone was largely due to the effect of microbial activities on lipid oxidation, where a number of *Penicillium* and *Aspergillus* species may degrade fatty acids with short and medium chain lengths to 2-heptanone [7]. Therefore, the possibility that molecular hydrogen could decrease bacterial growth (or their specific enzymatic activities) in rice grain during storage could not be easily ruled out, and the corresponding mechanisms require further study. Of note, in our previous study, after preharvest HNW application, the white rice was plumper and glossier compared with the control [22]. In this study, preharvest HNW application could decrease rice off-flavor generation during storage. Therefore, in terms of appearance and smell, rice treated with HNW in the preharvest stage could be more competitive for attracting consumers.

### 2.2. Alleviation in Lipid Oxidation and Increased Antioxidant Machinery in Stored Rice

During storage period, the storage lipids in rice grain such as triglycerides are gradually oxidized to glycerol and free fatty acids, where the fatty acid could be further oxidized to various volatile compounds [7,26]. This finally leads to the deterioration of grain flavor. It is confirmed that fatty acid value (FAV) and thiobarbituric acid-reactive substances (TBARS) could indicate and reflect this change in grain oxidative state. Lipoxygenases (LOXs) have been well known to be participated in the above process [7,26,27].

Rice grains typically contain three isozymes, lipoxygenase1 (LOX1), lipoxygenase2 (LOX2), and lipoxygenase3 (LOX3) [28]. After one year of storage, the FAV (Figure 3A) and TBARS content (Figure 3B) in brown rice irrigated with HNW during planting were significantly lower than the control samples. Similarly, LOX activity was observed to be reduced either (Figure 3C). Consistently, the transcriptional levels of LOX1, LOX2, and especially LOX3 showed the similar tendencies (Figure 3D,E).

A previous study has shown that the nutritional quality loss and short shelf life of brown rice was principally caused by LOXs-dependent lipid peroxidation [28]. Medium chain fatty aldehyde in some plants has been well known to be mediated by LOX-mediated cleavage of linoleic and linolenic acid [7]. In addition, it has been that observed that H_2_ appears to negatively regulate the LOX activity of *Medicago sativa* under cadmium stress [29]. Combined with the profiles of volatile compounds (Figure 2), the above results clearly indicated that the preharvest application of HNW might down-regulate the transcription of LOX gene and its activity, thus reducing the levels of lipid peroxidation (TBARS) and the corresponding products in the stored rice.

Free radicals of oxygen have often been proposed to be the main causative agents of lipid autoxidation, and antioxidant enzymes, especially SOD, appear to alleviate the process of lipid autoxidation initiation [12]. As shown in Figure 4A, under our storage conditions, the free radical scavenging rate of ABTS in brown rice after irrigation with HNW was obviously higher than that in the control. Consistently, the activities of SOD, CAT, POD, and APX in stored rice were increased by HNW irrigation in comparison with the control (Figure 4B–E).

Several recent studies have shown that, hydrogen gas modified atmosphere packaging could abolish the decrease of ABTS radical scavenging capacity in chicken egg [30]. Besides, 3% H_2_ treatment significantly stimulated the activities of SOD, POD, CAT, and APX in chives during cold storage [17]. A recent study proposed that Fe-porphyrin might be a redox-related biosensor of hydrogen molecule for scavenging of ROS [31]. Porphyrin ring is the active center of CAT, POD and APX [32,33,34]. This seems to explain that H_2_ could induce the activity of some antioxidant enzymes to regulate the intracellular ROS to maintain a relatively low level, and reduce the permeability of cell membrane [35,36]. It is well known that the above antioxidant enzymatic activities delay the oxidative damage of plant foods during storage, thus prolonging the storage period [37]. Here, preharvest application of molecular hydrogen did alleviate lipid autoxidation in rice during storage by improving the antioxidant capacity via enhancing the activities of antioxidant enzymes.

### 2.3. Preharvest HNW Treatment Maintains High Levels of Amino Acid Contents

The nutritional limitation of rice mainly comes from a deficiency of the essential amino acids in rice protein, particularly lysine (Lys) [28]. Based on the determination of hydrolysis essential amino acid composition, we observed that the total content of hydrolyzed amino acids in brown rice irrigated with HNW was 25.00% higher than that in the control after storage (Appendix A). Importantly, the contents of valine (Val), leucine (Leu), phenylalanine (Phe), histidine (His), and Lys in brown rice after HNW irrigation were significantly higher (Figure 5A). With the exception of proline (Pro) and cysteine (Cys), the contents of the nonessential amino acids in brown rice were maintained at relatively high levels after HNW treatment (Figure 5B). Therefore, we speculate that the preharvest application of HNW may change the proportion of amino acids in brown rice. Combining the results of the amino acid contents (Figure 5) with our previous discovery that molecular hydrogen application at preharvest could improve element content, such as iron, magnesium, etc. [22], we further propose that molecular hydrogen could improve nutritional values, which is beneficial for human health.

The taste quality of rice is the result of the interaction of many components, including protein, amylose, etc. [38]. In our study, there was no significant difference in protein content between the two treatments (Appendix A). At the same time, according to the previous study [39], the above results might be not correlated with the taste of rice. However, we found that HNW could significantly reduce the amylose/starch ratio of stored rice compared to the control, and this is consistent with our previous research [22] (Appendix A), which was beneficial for improving the taste quality of stored rice [40]. In order to understand the effects of molecular hydrogen on the taste quality of stored rice, further exploration is required.

### 2.4. The Influence of HNW on Rice Storage Characters by Correlation Analysis and PLS-DA

During storage, the free fatty acid value (FAV) of rice was significantly increased [27]. In addition, the volatile compounds of lipid autoxidation and LOX-enzymatic oxidation were increased during storage, thus leading to the off-flavor of the rice [4]. Subsequent regression analysis clearly showed that the characteristic volatiles contents in this study have a strong positive correlation with FAV, LOX activity, and the transcriptional levels of LOX1, LOX2, and LOX3, but they have negative correlations with ABTS free radical scavenging and the activities of antioxidant enzymes, which is consistent with the conclusions of previous studies [9,12,24] (Figure 6A).

PLS-DA analysis shows that two components could explain 59.8% (component 1) and 20.3% (component 2) of the total variance (Figure 6B). The HNW treatment and the control group were clearly separated by component 1. 

A VIP analysis of the PLS-DA determined the key characteristics participating in the differentiation between the HNW treatment and the control groups (VIP scores > 1), including the characteristic volatiles, ABTS free radical scavenging activity, four antioxidant enzyme activities, LOX activity and three transcripts, and FAV (Figure 6C).

Additionally, the separation of the HNW treatment and the control clusters is observed in Figure 6D (the clustering dendrogram). The above results further indicate that preharvest HNW treatment may improve the storage quality of rice, and a number of recent studies have shown that H_2_-modified atmosphere packaging could extend the storage quality of chicken egg [30] and postharvest Chinese chives [17]. Therefore, it can be suggested that molecular hydrogen can be applied in both the preharvest and postharvest periods compared to the application of the N_2_-modified atmosphere or oxygen-absorbing material used only for food postharvest storage. We further propose that the application of molecular hydrogen during manufacturing and transportation might be beneficial for the improvement of postharvest storage quality and the nutritional values of agriculture products, including cereal-based foods.

## 3. Materials and Methods

### 3.1. Plant Material and Treatments

According to the previous report [38], the field management of the rice (*Oryza sativa* ‘Huruan1212’) took place in Jurong, Jiangsu Province, China, and it includes the time and volume of irrigation with the ditch water (Con) and the hydrogen nanobubble water (HNW; injecting the hydrogen nanobubbles into the ditch water) treatment groups, as well as the determination of the H_2_ concentration in the HNW (Appendix A), that was carried out in 2020 [22]. Under our experimental conditions, the content of H_2_ in the machine outlet was approximately 0.5 mM (1000 ppb) and the final concentration of H_2_ in rice field after HNW treatment was approximately 0.05 mM (100 ppb, Appendix A). Further, the half-life of H_2_ in the HNW was more than 3 h [22].

On 1 November 2020, after cleaning and air drying, 1 kg harvested rice grains for each treatment group were respectively put into nylon net bags. The rice grain was stored in an incubator at constant temperature of 25 °C, 70% RH, in laboratory conditions at Nanjing Agriculture University until 31 October 2021. After one year of storage, from each treatment, 300 g rice (moisture content of approximately 11–13%; Appendix A) grain was hulled and 60 g rice, with complete embryos, were randomly selected for one biological repetition and ground into flour by an A11 basic grinder (IKA, Staufen, Germany), with or without a liquid nitrogen condition. There were three biological repetitions for each treatment. The samples were sieved through an 80-mesh sieve (178 μm) and then stored at −80 °C for analysis. 

### 3.2. Extraction and Analysis of Volatiles

The volatiles were collected by headspace solid-phase microextraction (SPME) [25]. The samples (3 g) were used for determination with the addition of 2,4,6-trimethylpyridine (TMP, equivalent 1 μg, internal standard, Macklin, Shanghai, China) in a vial. The volatiles were extracted with SPME fiber (DVB/CAR/PDMS Stable Flex, Supelco, Burlington, MA, USA) at 70 °C for 30 min and analyzed by a gas chromatography-mass spectrometry system (Intuvo 9000 GC and 7000D MS; Agilent Technologies, Santa Clara, CA, USA) with an Agilent HP-5 column. The flow rate of the carrier gas (helium, 99.999%) was set to 1 mL min^-1^. The injection temperature was 250 °C, in the splitless mode [41]. The oven temperature was held at 40 °C for 3 minutes, and then it was increased by 10 °C min^−1^ to a final temperature of 230 °C and maintained for 5 min. The transfer line and ion source were 280 °C and 230 °C. EI ionization mass spectra were determined at 70 eV, and the scan range was set to 33–550 amu.

The identification of volatiles used combined mass spectrometry data with the NIST 17 standard library data to determine the relative content of the compounds, with reference to the internal standard (μg kg^−1^) [25,42].

### 3.3. Quantification of Fatty Acid Value (FAV) and Thiobarbituric Acid Reactive Substances (TBARS)

The determination of FAVs was carried out according to Kechkin et al. [26]. The values were expressed by the mass (mg) of NaOH required to neutralize the free fatty acids (pH 7.0) contained in 100 g of rice flour.

The contents of the TBARS were analyzed by the colorimetric method [17] to evaluate the lipid peroxidation. The content of the TBARS was expressed as μmol kg^−1^ fresh weight.

### 3.4. Determination of 2,2′-azino-bis(3-ethylbenzothiazoline-6-sulphonic Acid (ABTS) Radical Scavenging Rate

The ABTS radical scavenging rate was determined according to the previous method [30].

### 3.5. Assay of Antioxidant Enzymes and Lipoxygenase (LOX) Activities

The antioxidant enzyme activity in the brown rice flour was estimated by following the methods described in previous reports [17]. The enzyme unit (U) of SOD is defined as the amount of enzyme required by the inhibition of 50% of the nitro blue tetrazolium reduction. The activity of catalase (CAT) was determined by measuring the reduction in hydrogen peroxide (H_2_O_2_) at 240 nm. Peroxidase (POD) activity was measured by monitoring the remission of guaiacol oxidation at 470 nm. Ascorbate peroxidase (APX) was analyzed by monitoring the decrease of 1 mM (*w/v*) ascorbic acid at 290 nm.

Lipoxygenase activity in the brown rice flour was determined based on the spectrometric method [43]. The increase in absorbance value at 234 nm per mg of protein by 0.001 per min was used as the enzyme activity unit (U). A BCA protein quantification kit was used for protein quantification (Vazyme, Nanjing, China).

### 3.6. Transcriptional Analysis of Lipoxygenase Genes

The total RNA in brown rice was extracted using the CTAB-LiCl method [44]. The determination of RNA concentration and quality, cDNA synthesis, and qPCR were referenced by 2^−ΔΔCT^ [25]. *OsActin* and *OsUBIQUITIN* [45,46] were the reference genes. The primers are shown in Appendix A.

### 3.7. Analysis of Hydrolysis Amino Acid Content

The hydrolyzed amino acids were analyzed by a Sykam S433D amino acid analyzer (Sykam, Munich, Germany) with a column packed with LCA K07/Li. For each treatment group, the brown rice flour from three replicates was mixed and set as the mixed samples. The extraction and analysis were completed according to Weiss et al. [47]. After acid hydrolysis (6 M hydrochloric acid), quantitation was carried out using an external standard (17 amino acid standards, Yuanye, Shanghai, China) method.

### 3.8. Statistical Analysis

The values are the means ± standard deviations (SD) of the three biological repetitions. Differences among the treatments were analyzed by t-test using SPSS 26.0 (IBM SPSS Inc., Chicago, IL, USA), and a *p*-value of <0.05 was considered statistically significant.

The characteristic volatiles contents, FAV, ABTS radical scavenging rate, TBARS, LOX activity, transcripts of LOX genes and antioxidant activity, and changes in essential amino acids were considered as storage characteristics. Using MetaboAnalyst 5.0 (https://www.metaboanalyst.ca, accessed on 31 May 2022), the data of the above storage characteristics were analyzed with correlation analysis, partial least squares-discriminant analysis (PLS-DA), and hierarchical clustering analysis (HCA).

## 4. Conclusions

Maintenance of the quality of stored rice is vital for its consumer acceptability. This study showed that the preharvest HNW treatment exerted positive effects on maintaining the flavor and amino acid nutrition of rice during storage, which might be attributed to the increased antioxidant capacity and reduced lipid oxidation through regulating LOX activity and transcripts. Therefore, this trial showed a great potential for molecular hydrogen preharvest application for rice during the postharvest storage stage.

## Figures and Tables

**Figure 1 plants-11-02588-f001:**
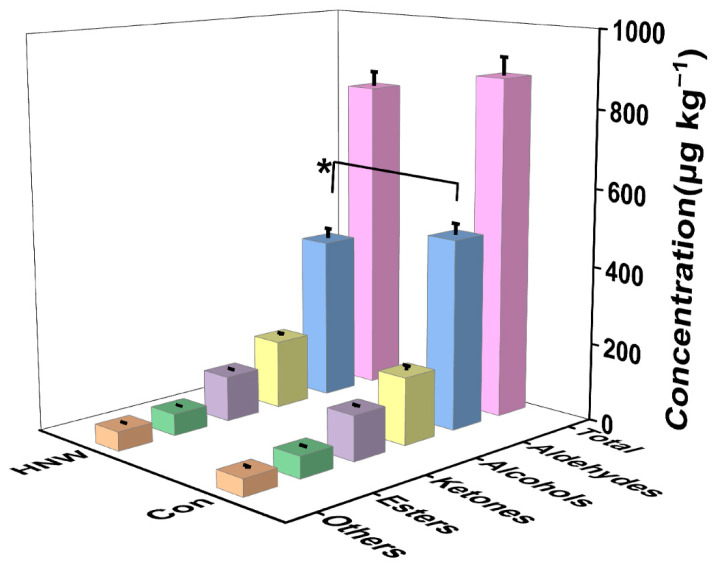
The volatile profiles of one year stored rice ‘Huruan 1212’ irrigated with ditch water (Con) or HNW during rice planting. Concentrations were expressed as μg kg^−1^ of rice fresh weight, equivalent to 2,4,6-Trimethylpyridine (TMP). Values were mean ± SD. * indicates significant difference (*t*-test, *p* < 0.05).

**Figure 2 plants-11-02588-f002:**
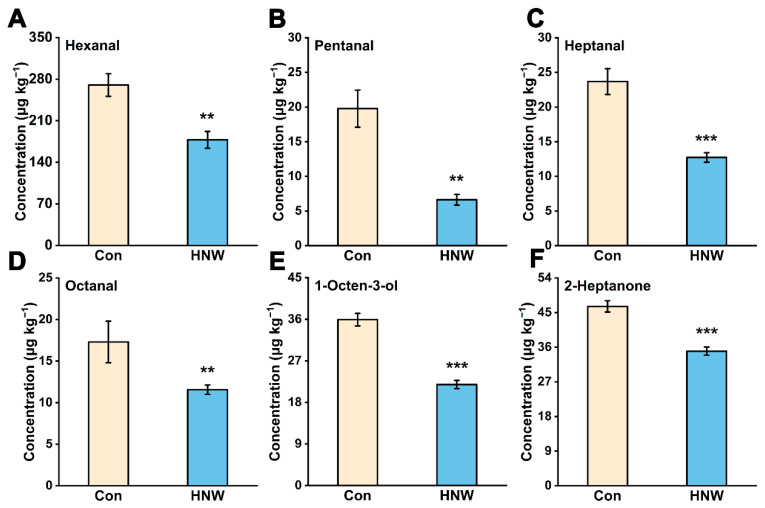
Changes in (**A**) hexanal, (**B**) pentanal, (**C**) heptanal, (**D**) octanal, (**E**) 1-octen-3-ol, and (**F**) 2-heptanone contents after one year of storage. Concentrations were expressed as μg kg^−1^ of rice fresh weight. Values were mean ± SD. **, *** indicate significant difference (*t*-test, *p* < 0.01 or 0.001).

**Figure 3 plants-11-02588-f003:**
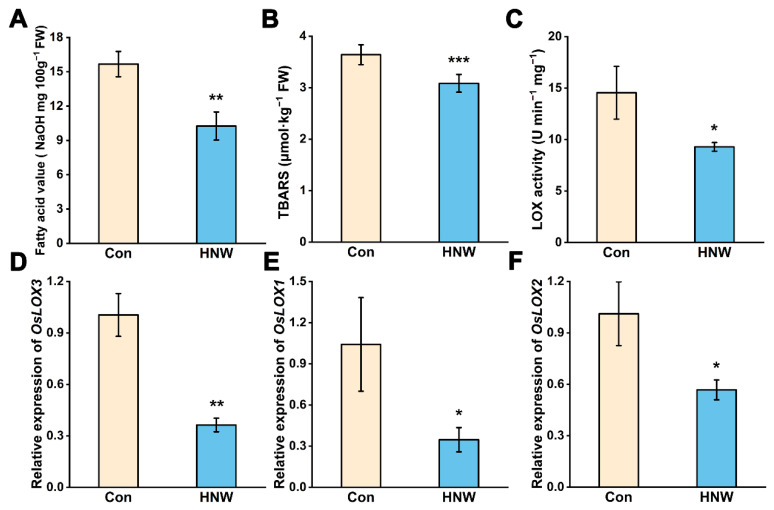
Profiles of (**A**) fatty acid value, (**B**) TBARS, (**C**) LOX activity, and (**D**–**F**) their transcripts (OsLOX-3/1/2) after one year of storage. Values were mean ± SD. *, **, *** indicate significant difference (*t*-test, *p* < 0.05, 0.01, or 0.001).

**Figure 4 plants-11-02588-f004:**
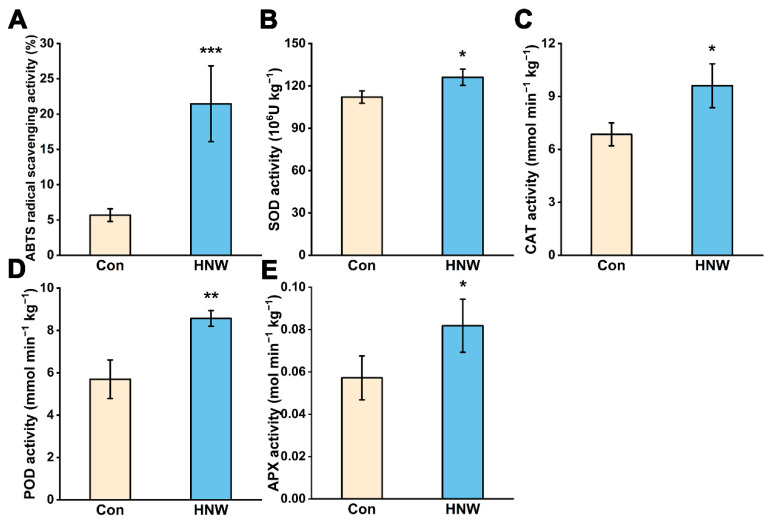
Changes in (**A**) ABTS radical scavenging rate and enzymatic activity of (**B**) SOD, (**C**) CAT, (**D**) POD, (**E**) APX after one year of storage. Values were mean ± SD. *, **, *** indicate significant difference (*t*-test, *p* < 0.05, 0.01, or 0.001).

**Figure 5 plants-11-02588-f005:**
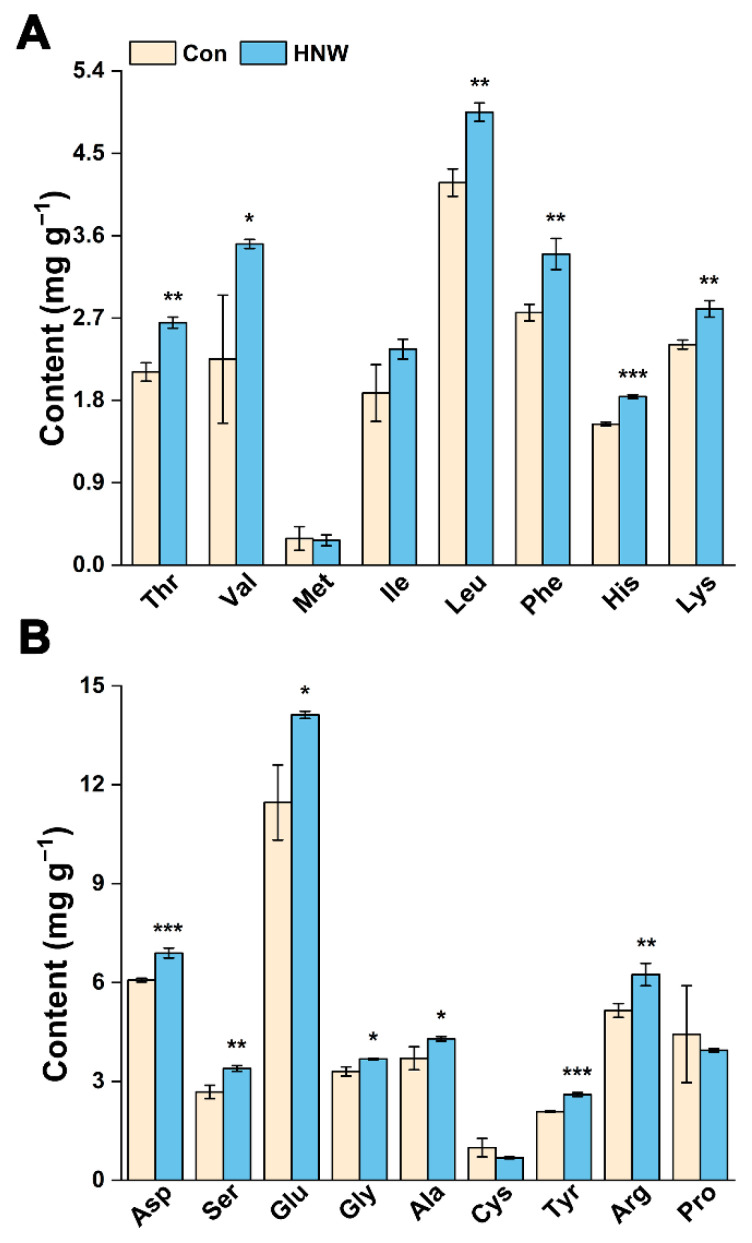
Changes in (**A**) essential and (**B**) nonessential amino acids contents after one year of storage. Values were mean ± SD. *, **, *** indicate significant difference (*t*-test, *p* < 0.05, 0.01, or 0.001).

**Figure 6 plants-11-02588-f006:**
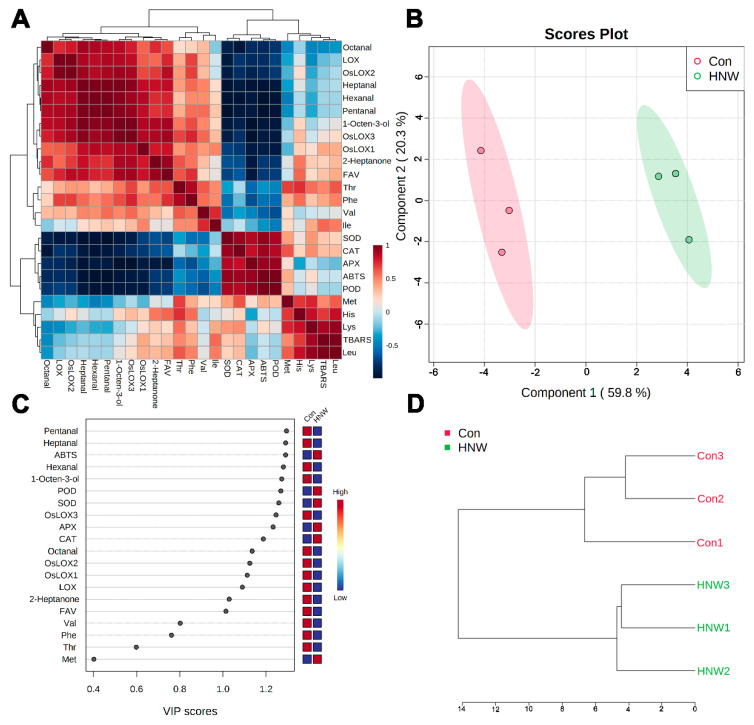
Comprehensive multivariate analysis of storage characteristics of one year stored rice irrigated with ditch water (Con) or HNW during rice planting. (**A**) Correlation heat map of storage characteristics. (**B**) Score plot of PLS-DA. (**C**) VIP scores of PLS-DA. The colored boxes indicate the relative concentrations. (**D**) Dendrogram of samples based on HCA.

## Data Availability

Not applicable.

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
