# Peer review of "Molecular Hydrogen Improves Rice Storage Quality via Alleviating Lipid Deterioration and Maintaining Nutritional Values"

_plants, 2022, doi:10.3390/plants11192588_

Round 1

Reviewer 1 Report

Research showing that molecular hydrogen improves rice quality looks interesting. However, I have a few questions about the experimental design.

First, why was the experiment conducted with only one treatment group?
The result shows the differences between control and 0.5mM H2 treatement. If it was compared with other concentration treatments, for example, 0.1mM, 0.25mM, or 1mM treatment, it seems that a more accurate interpretation of the H2 treatement effect was possible.

Second, if there is data measuring the nutritional content of rice before storage, I would like to add it in manuscript or supplementary data. Since they were cultivated separately, I think the comparison before and after storage is also meaningful.

And other points;

Line 221. How was the relative humidity of the incubator?

Line 200 - 205.
'low-carbon society'

How much energy is needed to artificially synthesize hydrogen? 

This part seems to need additional explanation or deletion.

Reviewer 2 Report

This is an interesting paper reporting an application of hydrogen nanobubble water (HNW) for rice agriculture. Antioxidative functions of hydrogen gas (molecular H2) as well as HNW have been reported to be effective to improve human health. This study evaluated its effectiveness on rice storage in terms of its nutritional value. The data presented clearly suggest that the HNW treatment gives positive effects on suppression of oxidation of fatty acids accompanying with amino acid composition changes. I think that the results are worthy of publication. Before its publication, the authors are requested to consider the following points.

The biochemical analyses beautifully confirm positive effects of HNW on rice nutritional quality in its storage. From a biochemical point of view, I can agree with most of the statements described in this manuscript. My concern is about background issues of this study as the below.

1) Lipoxygenase

In this study, lipoxygenase as well as other antioxidant enzymes are argued. In general, an enzymatic activity requires sufficient solvent water, a condition is not good for storage of rice. In my understanding, the rice grains are dried with hot air before their storage, a condition that denatures proteins to inactivate enzymes. I understand that nonenzymatic or chemical auto-oxidation should occur in ambient conditions but wonder why enzymatic reactions proceeded even during the storage. 

2 ) Ditch water

According to Materials and Methods, the authors compared the biochemical parameters between HNW and ditch water treatment groups (line 214). In general, water quality for irrigation is very critical for cultivating a high quality of rice. Since there is no description on the water quality of the ditch water, I feel difficulties to assess the HNW effects. There may be two possibilities: either “direct” positive effects on rice quality or “indirect” suppressive effects on negative factors or components contained in ditch water such as calcium ions, heavy metals or microbial association (line 106). It would be helpful, if the authors could mention the water quality of the ditch water used for control experiments. Note that H2 gas may be contained in wet soils due to microbial activities. We may have overlooked contributions of such natural H2 gas production which potentially varies among water quality and soil compositions.  

3) Antioxidative functions

If this paper would be submitted as application science, the above are my concerns. If the paper is submitted as basic science, I need to say that it is weak in discussion in the current manuscript. If oxidative reactions during the storage is the sole reason of lipid deterioration and generation of the off-flavors in stored rice, one may consider the storage of rice with cheap N2 gas or O2 absorbing materials to remove molecular oxygen. This strategy has been confirmed in food science. It would be wonderful, if the authors could discuss more about how H2 gas influences multiple biochemical processes with citing previous literatures published in health and food sciences.

Reviewer 3 Report

  1. Include reference for the Method used for the "Extraction and analysis of volatiles" section

Reviewer 4 Report

1. We know that the protein content of rice is negatively correlated with the eating value of rice. Data on the protein content of rice after the new treatment should be given.

2. The change of amino acid composition will also cause the change of rice taste value. If the taste value decreases significantly, the value of rice will be reduced, which may lead to the embarrassment of increasing yield without increasing income. Therefore, it is suggested to give the data of rice taste value.

Round 2

Reviewer 1 Report

The manuscript was appropriately revised.

Author Response

Thank you

Reviewer 4 Report

The authors have made a reasonable explanation.

Author Response

Thank you